# Negri bodies are viral factories with properties of liquid organelles

Jovan Nikolic [1], Romain Le Bars [1], Zoé Lama[1], Nathalie Scrima[1], Cécile Lagaudrière-Gesbert[1], Yves Gaudin [1] & Danielle Blondel[1]

Replication of *Mononegavirales* occurs in viral factories which form inclusions in the host-cell cytoplasm. For rabies virus, those inclusions are called Negri bodies (NBs). We report that NBs have characteristics similar to those of liquid organelles: they are spherical, they fuse to form larger structures, and they disappear upon hypotonic shock. Their liquid phase is confirmed by FRAP experiments. Live-cell imaging indicates that viral nucleocapsids are ejected from NBs and transported along microtubules to form either new virions or secondary viral factories. Coexpression of rabies virus N and P proteins results in cytoplasmic inclusions recapitulating NBs properties. This minimal system reveals that an intrinsically disordered domain and the dimerization domain of P are essential for Negri bodies-like structures formation. We suggest that formation of liquid viral factories by phase separation is common among *Mononegavirales* and allows specific recruitment and concentration of viral proteins but also the escape to cellular antiviral response.

[1] Institute for Integrative Biology of the Cell (I2BC), CEA, CNRS, Univ. Paris-Sud, Université Paris-Saclay, 91198 Gif-sur-Yvette cedex, France. Correspondence and requests for materials should be addressed to Y.G. (email: yves.gaudin@i2bc.paris-saclay.fr) or to D.B. (email: danielle.blondel@i2bc.paris-saclay.fr)

Replication and assembly of many viruses occur in specialized intracellular compartments known as viral factories, viral inclusions or viroplasms[1, 2]. These neo-organelles formed during viral infection concentrate viral proteins, cellular factors and nucleic acids to build a platform facilitating viral replication. They might also prevent the activation of host innate immunity and restrain the access of viral machineries to cellular antiviral proteins. Such factories are widespread in the viral word and have been identified for a variety of non-related viruses.

The location and the nature of viral factories are very heterogeneous. They depend on the genome composition (DNA or RNA) and on the viral replication strategy. The first viral factories which were characterized were those formed by large DNA viruses such as the Poxviridae, the Iridoviridae and the Asfaviridae[3–6]. Those factories are devoid of membrane, located in close proximity to the microtubule organizing center. They recruit mitochondria, contain molecular chaperones such as HSP proteins and are surrounded by a vimentin cage. In the case of positive strand RNA viruses, viral factories are associated with rearrangements of membranes from diverse organelles (Mitochondria, ER, and so on) leading to the formation of double-membrane vesicles[7–9]. These vesicles seem to remain connected to the cytoplasm by channels which allow ribonucleotide import and product RNA export.

Several negative strand RNA viruses also induced the formation of membrane-less cytoplasmic inclusions which, in the case of rhabdoviruses[10, 11] and filoviruses[12], have been demonstrated to harbor several viral replication stages. In the case of rabies virus (RABV), those inclusions are called Negri bodies (NBs) and can reach several microns in diameter[11, 13, 14].

RABV (Mononegavirales order, Rhabdoviridae family, Lyssavirus genus) is a neurotropic virus, which remains a substantial health concern as it causes fatal encephalitis in humans and animals and still kills > 55,000 people worldwide every year mainly in Asia and Africa. It has a negative stranded RNA genome (about 12 kb) encoding five proteins. The genome is encapsidated by the nucleoprotein (N) to form a helical nucleocapsid in which each N protomer binds to nine nucleotides[15]. The nucleocapsid is associated with the RNA dependent RNA polymerase (L) and its cofactor the phosphoprotein (P) to form the ribonucleoprotein (RNP) which is enwrapped by a lipid bilayer derived from a host cell membrane during the budding process. The matrix protein M and the glycoprotein G are membrane-associated proteins. M protein is located beneath the viral membrane and bridges the condensed RNP and the lipid bilayer. G protein is an integral transmembrane protein that is involved in viral entry[16]. The virus enters the host cell through the endocytic pathway via a low-pH-induced membrane fusion process catalyzed by G. The RNP is then released into the cytoplasm and serves as a template for transcription and replication processes that are catalyzed by the L–P polymerase complex. During transcription, a positive-stranded leader RNA and five capped and polyadenylated mRNAs are synthesized. The replication process yields nucleocapsids containing full-length antigenome-sense RNA, which in turn serve as templates for the synthesis of genome-sense RNA. Replication strictly depends upon ongoing protein synthesis to provide the N protein necessary to encapsidate nascent antigenomes and genomes. Neo-synthesized genomes either serve as templates for secondary transcription or are condensed and assembled with M proteins to allow budding of neo-synthesized virions at a cellular membrane[16].

We have previously demonstrated that viral transcription and replication take place within NBs[11]. NBs contain all the replication machinery (L, N and P)[11] together with M[17] and several cellular proteins including HSP70[18] and the focal adhesion kinase (FAK)[19]. It has been recently shown that NBs are in close proximity to stress granules (SGs)[20], which are membrane-less liquid cellular organelles consisting of mRNA and protein aggregates[21, 22] that form rapidly in response to a wide range of environmental cellular stresses and viral infections[23].

In this report, using a recombinant RABV expressing a fluorescent P protein, we have investigated the physical nature of NBs. Live imaging and FRAP analysis demonstrated that they have liquid organelles properties. We have also characterized the role of the cytoskeleton on the dynamics of NBs and transport of RNPs. This revealed that RNPs are ejected from NBs by a cytoskeleton-independent mechanism and are further transported along microtubules. Finally, we developed a minimal system which recapitulates NBs properties and allows us to identify the P domains required for NBs formation.

## Results

**Characterization of inclusions formed during RABV infection.** RABV-infected BSR cells (MOI = 0.5) were fixed and permeabilized at different times post infection (p.i.), and the structures formed in the cytosol were analyzed by immunofluorescence with an anti-N antibody (Fig. 1a). At each time, the structures were counted and classified based on their size (i.e., the surface of their projection) (Figs. 1b–d). We distinguished small dots (surface < 0.26 μm²) (Fig. 1b), inclusions of intermediate size (0.26 μm² < surface < 3.7 μm²) (Fig. 1c) and large inclusions (surface > 3.7 μm²) (Fig. 1d). At early time p.i. (Fig. 1a,c, 8 h), there is a limited number (up to 2 per cell) of spherical intracytoplasmic inclusions of intermediate size, which correspond to the first NBs formed during the infection[11, 20]. These structures grow with time (Fig. 1a,e) and sometimes lose their spherical aspect (Fig. 1a, 24 h). At later stages of infection (16 h and 24 h), large inclusions (1–3 per cell in most cases) are observed together with an increased number of intermediate inclusions and an explosion of the number of punctate structures (Fig. 1a–d).

The cellular location of inclusions and punctate structures was analyzed by confocal microscopy. At 24 h p.i., punctate structures are mainly detected in the basal section, whereas inclusions (intermediate and big ones) are located in median sections of the cell (Fig. 1f).

This was confirmed by electron microscopy performed on fixed sample (Fig. 1g,h). Cytoplasmic inclusions are rather observed in median section (Fig. 1h). They are initially highly spherical and devoid of membranes (Fig. 1h, 8 h). At later stage (Fig. 1h, 24 h), they are often associated with membranes—most probably derived from the ER[11] – and their shape appears to be altered. At that time, basal section of the cells reveals the presence of typical condensed RNPs (Fig. 1g). Therefore, the small punctate structures observed by immunofluorescence most probably correspond to condensed RNPs. The inclusions, either large or intermediate, will be referred as NBs thereafter.

**NBs are liquid structures.** To gain insight into the dynamic of NBs inside the cytoplasm, live cell imaging was performed. For this, BSR cells were infected with the previously described recombinant virus rCVSN2C-P-mCherry[20], which expresses the P protein C-terminally fused to the mCherry fluorescent protein. This recombinant virus behaves as wild type and, particularly, exhibits similar kinetics of NBs formation (Supplementary Fig. 1).

Up to 16 h p.i., NBs appear to be highly spherical structures (Figs. 1a,h) as judged by the distribution of their axial ratio which is < 1.2 in 70% of cases (Fig. 2a). This suggests that they consist of viscous liquids, similarly to the liquid-like nature of P granules[24, 25], SGs[21, 22] and nuclear bodies[26]. Consistent with this view, we frequently observed that when two or more NBs

contact one another, they readily fuse and round up into a single-larger sphere (Fig. 2b, Supplementary Movies 1 and 2). Furthermore, we also sometimes observed spherical bubbles crossing the NBs (Fig. 2c, Supplementary Movie 3). The directionality of their movement suggests that they are vesicles being trafficked through the NBs. This reinforces the idea that

NBs are made of a fluid phase, which can reversibly deform when encountering a physical barrier.

It has been reported that liquid organelles respond rapidly to cellular osmotic shock[27]. To test whether NBs also behave in this way, BSR cells that have been infected with rCVSN2C-P-mCherry for 16 h, were incubated with DMEM medium diluted 5 times in

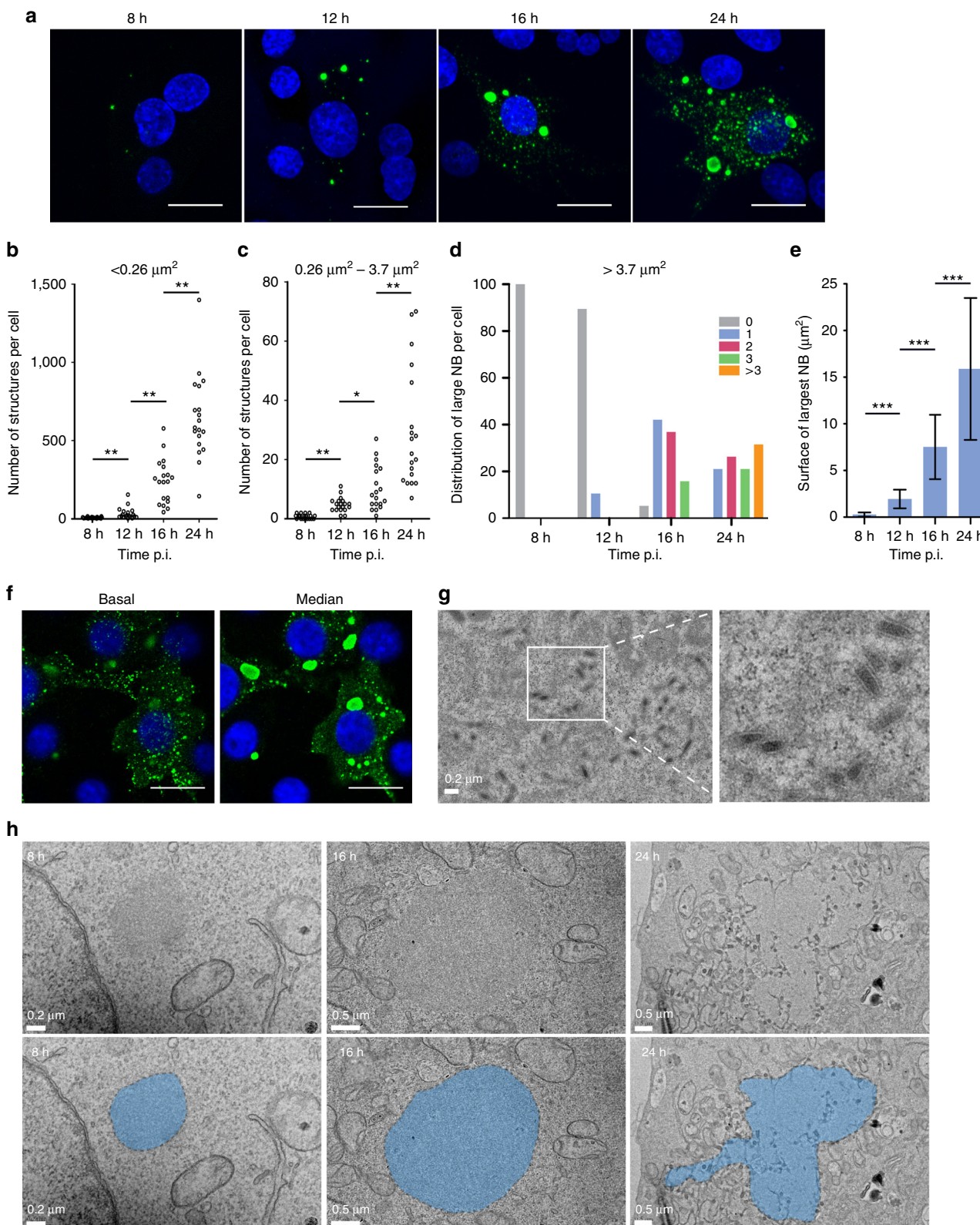

water. This hypotonic shock resulted in a fast and complete disappearance of NBs followed by the reformation of inclusions of intermediate size in 10–15 min (Fig. 2d, Supplementary Movie 4).

To definitively demonstrate the fluid nature of NBs, we performed fluorescence recovery after photobleaching (FRAP) experiments. For safety reason, we used a G gene deleted recombinant RABV (rCVSN2CΔG-P-mCherry), which was pseudotyped with the RABV G glycoprotein. This recombinant RABV also induces the formation of NBs, which incidentally demonstrates that G synthesis is not required for NB formation.

FRAP experiments were first performed on BSR cells transfected with a plasmid expressing P-mCherry (Fig. 2e and Supplementary Fig. 2). P-mCherry expressed alone was diffuse in the cytosol (Supplementary Fig. 2) and fluorescence recovery was fast although biphasic (Supplementary Table 1). The half time of the first phase for a photobleached region of 2.7 μm diameter took about 0.4 s corresponding to an approximate diffusion coefficient of $5 \times 10^{-12}$ m$^2$ s$^{-1}$ for P-mCherry in the cytosol (Supplementary Table 1), a value which is consistent with that expected for a protein of such a molecular weight. Full fluorescence recovery was achieved after ~30 s. Similar data were obtained in cells infected by rCVSN2CΔG-P-mCherry, when cytosolic P (i.e., outside the NBs) was photobleached (Fig. 2f and Supplementary Fig. 3).

When P located inside NBs was photobleached (Fig. 2g and Supplementary Fig. 4), the recovery of the fluorescence signal was again biphasic but much slower (Supplementary Table 1). The half time of the first phase for a photobleached region of 2.7 μm diameter is ~5.2 s (Fig. 2g), corresponding to a diffusion coefficient of $3.9 \times 10^{-13}$ m$^2$ s$^{-1}$ (Supplementary Table 1). Such a self-diffusion rate is consistent with those of other membrane-less organelles, such as nuclear speckles and nucleoli[21, 28]. After 2 min, the fluorescence recovery is not complete, but only reaches $78 \pm 8\%$ ($\pm$SD, $n = 12$). The recovery profile along a diameter revealed that fluorescence came back first to the periphery a few seconds after photobleaching and then displays a homogeneous distribution within the structure after ~1 min (Fig. 2h). Therefore, the FRAP data confirm that NBs have liquid properties and that P protein can shuttle between the cytosol and the inner of the NBs.

We have previously shown that SGs are formed in the cytoplasm of RABV infected cells[20]. As SGs have also been demonstrated to be liquid droplets[21, 22], we investigated if SGs and NBs can fuse. Cells were transfected by a plasmid encoding G3BP-GFP, which is a marker of SGs. Then, 1 h post transfection, they were infected with rCVSN2C-P-mCherry. As previously described, we observed SGs in close proximity to NBs (Fig. 3, Supplementary Movie 5). However, we never observed mixing content of the two structures: the P protein remains in NBs whereas G3BP remains in SGs. Furthermore, in some circumstances, we identified small SGs located inside NBs confirming the non-miscibility of the two liquid phases (Fig. 3).

**Role of the cytoskeleton on the NBs fate and RNPs transport.** To investigate the role of the cytoskeleton in NB morphogenesis and dynamics, we used several drugs which interfere with the organization of the cytoskeleton. Treatment of RABV infected cells with Nocodazole (NCZ), a drug which depolymerizes the microtubules, results in the disappearance of NBs of intermediate size (Fig. 4a,c) and a significant decrease in the number of small dots corresponding to condensed RNPs (Fig. 4a,b). In general, in the presence of NCZ, a single-large NB is observed (Fig. 4a,d) which is bigger than those observed in absence of the drug. This is evidenced by the significant increase of the average surface of the largest NB present in an infected cell (Fig. 4e). Taxol, which stabilizes the microtubule network, has no effect on the structures.

Cell treatment by Cytochalasin D (Cyto D), which inhibits actin polymerization, results in an increased fragmentation of NBs, which is apparent when considering the average surface of the largest NB present in an infected cell (Fig. 4e).

Live cell imaging was then performed in RABV infected cells. We observed ejections of punctate structures out of NBs (Fig. 5a, Supplementary Movies 6 and 7). These structures, which probably correspond to condensed RNPs, are rapidly transported further away across the cell (Supplementary Movie 8). Ejections are also observed when cells are treated by Nocodazole (Fig. 5b, Supplementary Movie 9). However, the ejected structures are not transported and remain in the vicinity of the NBs (Fig. 5b, Supplementary Movie 9).

We further characterized the transport of RNPs inside the cytoplasm (Fig. 5b). The histograms in Fig. 5c show that their velocity is the same in non-treated cells or in cells treated with Taxol or Cytochalasin D. However, in presence of Nocodazole, RNPs remain immobile in the vicinity of the NBs (Fig. 5b,c). Altogether with the fact that the measured velocities are in agreement with those of other cargoes for which transport is microtubule dependent, this indicated that RNPs transport requires the integrity of the microtubule network.

Indeed, when live imaging was performed on cells which have been transduced by a baculovirus expressing a tubulin fused with the GFP before infection by rCVSN2C-P-mCherry, we directly observed the transport of RNPs along the microtubules (Fig. 5d, Supplementary Movie 10).

**A minimal system recapitulating NB properties.** It has been previously demonstrated that co-expression of N and P after cell transfection also leads to the formation of cytoplasmic inclusions[29]. Indeed, in BSR cells constitutively expressing the T7 RNA polymerase (BSR-T7/5) and co-transfected by plasmids pTit-P and pTit-N, cytoplasmic spherical inclusions are observed (Fig. 6a).

The dependence of inclusions formation on the stoichiometry of the transfected plasmids was investigated. Spherical inclusions

**Fig. 1** Characterization of cytoplasmic inclusions in BSR cells infected by RABV. BSR cells were infected with CVS strain at a MOI of 0.5 and fixed at different times p.i. (8 h, 12 h, 16 h, 24 h). **a** Confocal analysis was performed after staining with a mouse monoclonal anti-N antibody followed by incubation with Alexa-488 donkey anti-mouse IgG. DAPI was used to stain the nuclei. Scale bars correspond to 15 μm. **b–d** Quantification of cytoplasmic structures labeled with anti-N antibody. At the indicated time p.i. the number of small dots (surface < 0.26 μm$^2$) **b**, of intermediate inclusions (surface between 0.26 and 3.7 μm$^2$) **c**, and of large inclusions (surface > 3.7 μm$^2$) **d** per cell was quantified using the image toolbox of MatLab software as described in the experimental procedures. *$p < 0.02$; **$p < 0.01$ ($n = 20$, two tailed Mann Whitney U test). **e** Average surface of the largest inclusion in the cell at the indicated time p.i. Surfaces of inclusions were determined using the image toolbox of MatLab software as described in the experimental procedures. The mean is shown with error bars representing the SD. ***$p < 10^{-4}$ ($n = 20$, two tailed Welch's t-test). **f** Confocal analysis revealing the basal localization of small dots and the median localization of inclusions in RABV infected cells at 24 h p.i. The analysis was performed after staining as in **a**. Scale bars correspond to 15 μm. **g, h** EM characterization of the ultrastructural aspects of BSR cells infected by RABV. Basal sections **g** reveal the presence of RNPs whereas median sections **h** reveal the presence of NBs displaying an electron dense granular structure (colored in blue in the bottom row) which lose their spherical shape when they associate with membranes at 24 h p.i

were observed with ratios of pTit-N and pTit-P plasmids going from 3:1 to 1:3. In limiting concentrations of one of the plasmids (ratios 9:1 or 1:9), no inclusions were observed even when both proteins were detected in the cell (Supplementary Fig. 6).

We investigated the physical properties of the inclusions formed in this minimal system. For this, we co-transfected

BSR-T7/5 cells with pTit-N and pTit-P-mCherry. We then performed FRAP experiments on pTit-P-mCherry located in the cytoplasmic inclusions that were formed. Recovery profiles (Fig. 6b and Supplementary Fig. 7), although less homogeneous, are however similar to those observed for P located in NBs in infected cells (Fig. 2g and Supplementary Table 1). This

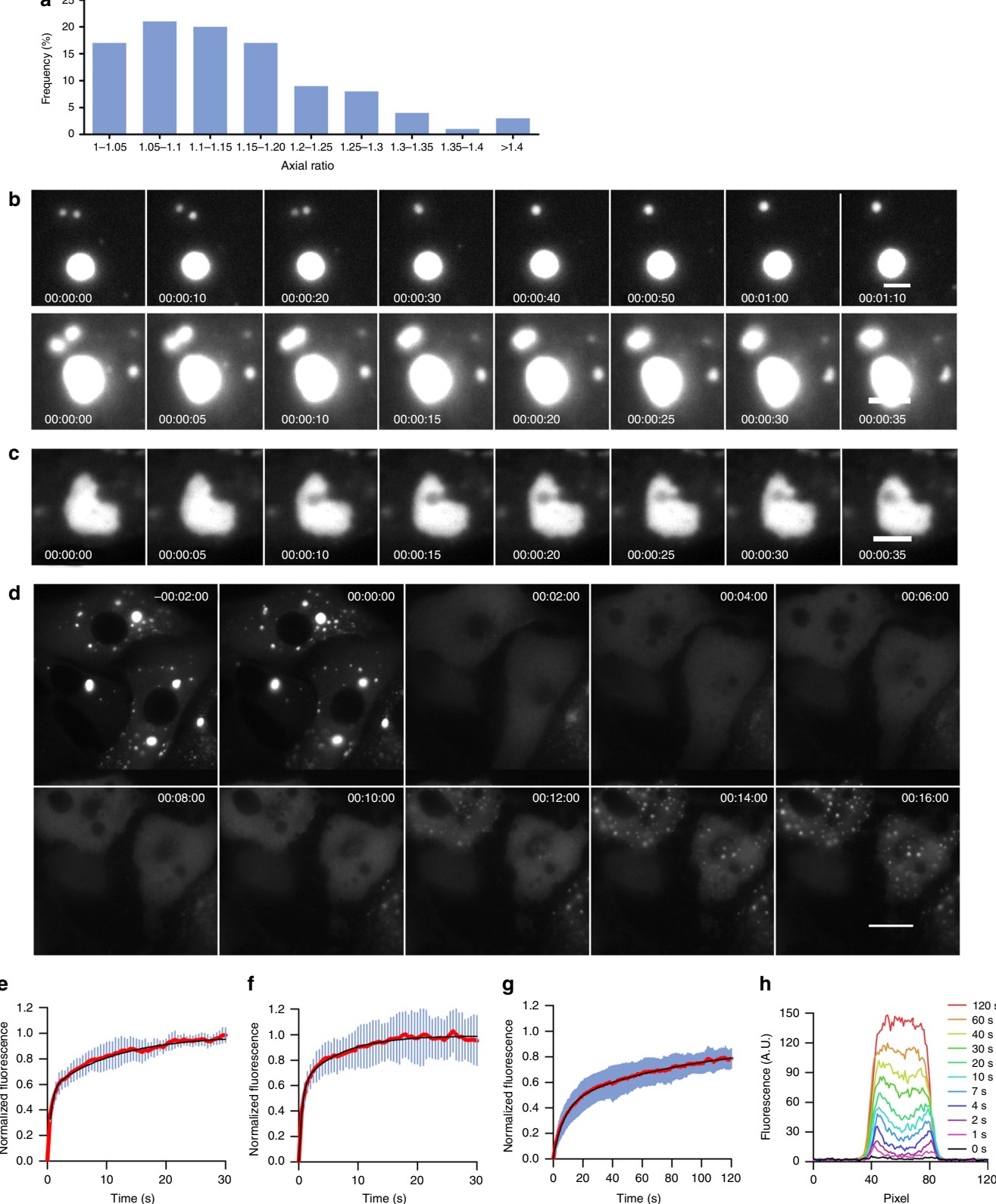

confirmed that the N-P inclusions formed in this minimal system have the same liquid characteristics as the NBs.

N-P inclusions also concentrate several cellular partners of P previously shown to be located in NBs such as HSP70 and FAK (Supplementary Fig. 8). However, unlike NBs, N-P inclusions do not eject material in the cytosol (Supplementary Movie 11, same images acquisition as Supplementary Movie 7). A possible role of RABV M protein in pinching off and ejection of RNPs was investigated. Co-tranfection of pTit-N and pTit-P-mCherry together with pTit-M-GFP (which allows expression of M-GFP) did not result in material ejection from N-P inclusions (Supplementary Movie 12, same images acquisition as Supplementary Movies 7 and 11).

We then took advantage of this minimal system to identify P domains that are essential for inclusions formation. P is a modular protein (Fig. 6c) containing an N-terminal domain (PNTD) which binds N0 (the soluble form of N protein devoid of RNA), a C-terminal domain (PCTD) which binds to the N associated with RNA, and two central intrinsically disordered domains (IDD1 and IDD2) flanking a dimerization domain (DD)[30]. Therefore, pTit plasmids allowing the expression of P deletion mutants were co-transfected with pTit-N and the presence of N-P inclusions was investigated. This revealed that the domains DD, IDD2 and PCTD are required for NB-like structures formation (Fig. 6d).

Domains IDD2 and PCTD are phosphorylated on residues S162, S210 and S271[31] (Fig. 6c). An eventual role of phosphorylation on the phase transition was investigated. Residues S162, S210 and S271 were mutated either into alanine residues (to abolish phosphorylation) or into aspartic residues (to mimic the phosphorylated state of the serine). None of those mutations affected P ability to form NB-like structures (Supplementary Fig. 9).

Finally, we delineated more precisely the part of IDD2 domain which is required for phase separation. Deletion of residues 151–181 did not affect P ability to form NB-like structures (Fig. 6e). Therefore, only the amino-terminal part of IDD2 (residues 132–150) is required for this process. This was confirmed by the deletion of residues 139–151, which abolished spherical inclusions formation (Fig. 6e).

## Discussion

In this study, we provide evidence that NBs are liquid droplets formed by phase separation. First, they form spherical structures (Figs. 1a,h and 2b) with a distribution of their axial ratios (Fig. 2a) which is very similar to that observed for other liquid organelles[32]. Second, they fuse together to form larger spherical structures (Fig. 2b). Third, they are crossed by spherical bubbles which are most probably vesicles being trafficked through their interior, showing that NBs can reversibly deform when encountering a physical barrier (Fig. 2c). Fourth, osmotic shock, induced by a rapid change from isotonic to hypotonic conditions, causes the rapid dissolution of NBs followed by the reformation of smaller spherical inclusions (Fig. 2d), a behavior which is reminiscent of that of the liquid cellular Ddx4-organelles[27]. Finally, the internal order within the NBs was assessed using FRAP measurements which definitively demonstrate their liquid nature (Fig. 2e–h).

Besides those physical properties, NBs share several other features with cellular liquid organelles[21, 25, 27, 33, 34]. Particularly, they are composed by viral RNA, an RNA binding protein (N), and a protein containing intrinsically disordered domains (P). Indeed, P and N expressed alone were able to form structures which recapitulate the properties of NBs (Fig. 6a). It is probable that in such N-P inclusions, N is associated with cellular RNAs and forms N-RNA rings and short RNP-like structures[35]. This minimal system allowed us to investigate the domains of P involved in this process. P dimerization domain, its N-RNA binding domain and the amino-terminal part of its second intrinsically disordered domain (IDD2) were absolutely required for NB-like structures formation (Fig. 6d). This could suggest that P dimers bridge RNPs to ensure the liquid phase cohesion. However, the P dimer is stable[36] and the affinity of P dimers for N-RNA rings (which mimic the structure of the viral RNPs) is quite high (Kd = 160 ± 20 nM)[37]. Such strong interactions are not compatible with a liquid behavior. Furthermore, in vitro experiments have indicated that P dimers are associated with N proteins of the same nucleocapsid[37]. Therefore, there must be other weak interactions either between P dimers or between nucleocapsids which remain to be characterized. We suggest that those weak interactions are mediated by the amino-terminal part of IDD2 as such intrinsically disordered domains have been involved in the formation of fuzzy complexes[38, 39].

The sequence of the amino-terminal part of IDD2 (residues 132–150, Supplementary Fig. 10) is strongly biased toward polar and charged residues which is a common feature of IDDs. It is also enriched in proline residues which, however, are not conserved among the lyssavirus genus. In fact, the IDD2 region (made of 51 residues) is not conserved except the two residues Q and T in position 147 and 148. This has to be compared with a 32% conservation of P amino acid sequence in the genus (Supplementary Fig. 10). Therefore, the ability of IDD2 to induce phase separation does not rely in its amino acid sequence but rather in some global physico-chemical properties which remain to be identified.

Fig. 2 NBs are liquid organelles. **a** NBs are close to spherical. Distribution of axial ratios of NBs observed 12 h p.i. (20 cells and 100 NBs). The axial ratio (a/b) was determined by fitting the NB to an ellipse having the same second moments of area using the image toolbox of MatLab software (a, b: long and short axes of the ellipse). **b** NBs can fuse together. BSR cells infected by rCVSN2C-P-mCherry were imaged at the indicated time (lower left corner). The initial time corresponds to 15 h p.i (top row) and 18 h p.i. (bottom row). Images have been extracted from Supplementary Movies 1 and 2 and are shown at 10-sec intervals (top row) and 5-sec intervals (bottom row). Scale bars: 3 μm. **c** Spherical bubbles cross NBs. BSR cells infected by rCVSN2C-P-mCherry were imaged at the time indicated (lower left corner). The initial time corresponds to 16 h p.i. Images have been extracted from Supplementary Movie 3 and are shown at 5-s intervals. Scale bar: 3 μm. **d** NBs are sensitive to a hypotonic shock. BSR cells infected by rCVSN2C-P-mCherry were imaged. A hypotonic shock was applied at indicated t = 0 (corresponding to 18 h p.i.). Images are shown at 2-min intervals. Images have been extracted from Supplementary Movie 4. Scale bar: 15 μm. **e–h** Fluorescence recovery after photobleaching (FRAP) of P-mCherry in BSR cells at 37 °C. The diameter of the photobleached regions was 2.7 μm. **e–g** FRAP data were corrected for background, normalized to the minimum and maximum intensity. The mean is shown with error bars representing the SD. Experimental curves were fitted with a two-exponential model (in black). **e** Cytosolic P-mCherry expressed in BSR-T7/5 cells was photobleached 24 h after transfection of pTit-P-mCherry plasmid. Data were from 11 FRAP events (Supplementary Fig. 2). **f** Cytosolic P-mCherry expressed in BSR cells infected by rCVSN2CΔG-P-mCherry was photobleached 16 h p.i. Data were from 13 FRAP events (Supplementary Fig. 3). **g** P-mCherry localized in NBs in BSR cells infected by rCVSN2CΔG-P-mCherry was photobleached 16 h p.i. Data were from 12 FRAP events (Supplementary Fig. 4). **h** Fluorescence recovery profile along a diameter of a photobleached NB as in **g**

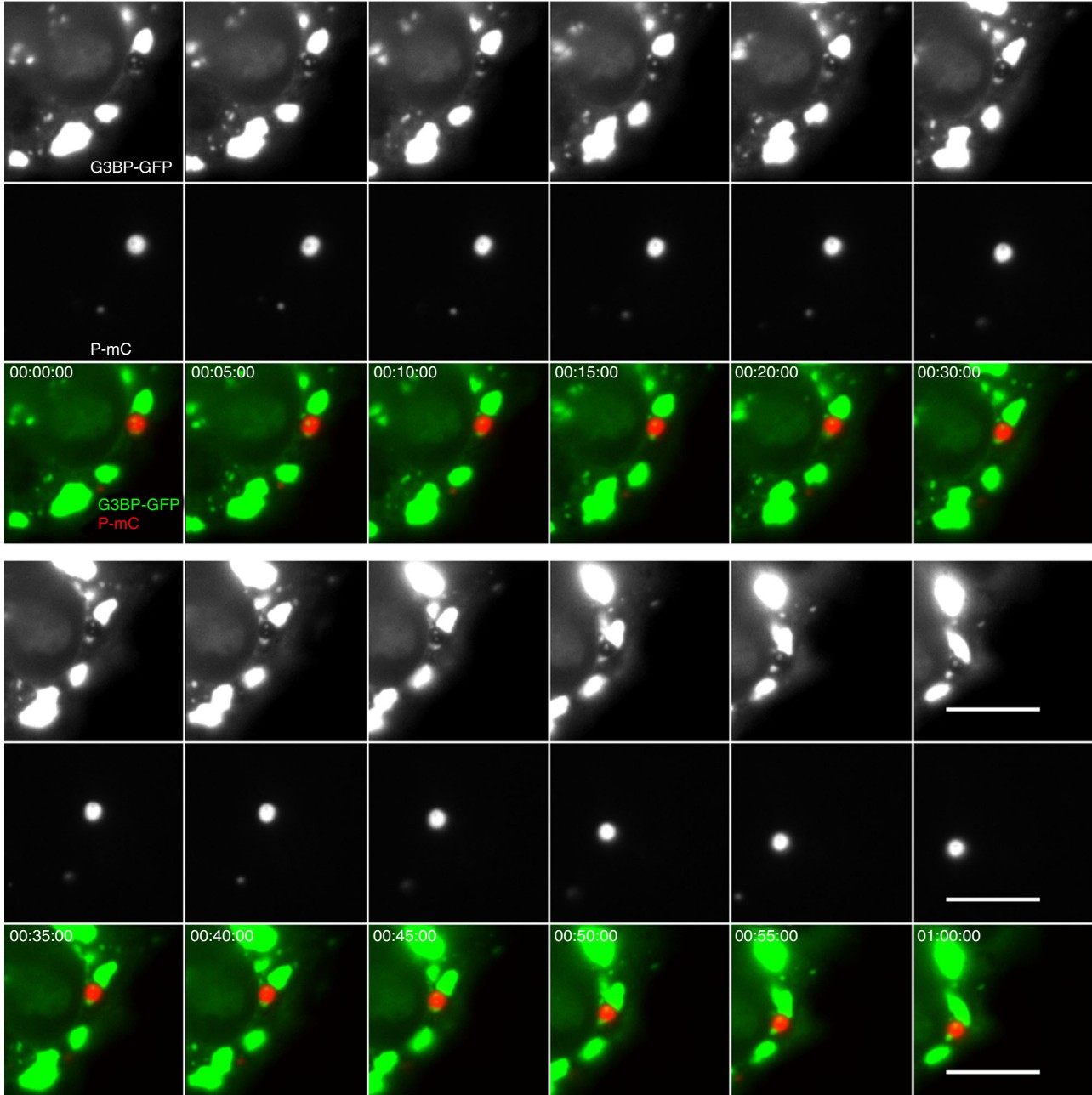

**Fig. 3** NBs and SGs are non-miscible liquid organelles. U373-MG cells were transiently transfected with pG3BP-eGFP (to visualize SGs, *top row*). 1 h post transfection, they were infected with the recombinant virus rCVSN2C-P-mCherry (*medium row*). Live-cell time-lapse experiments were performed at 16 h p. i. G3BP-GFP signals (*green*) and P-mCherry signals (*red*) are merged in the *bottom row*. The time post-infection is displayed in the *upper left corner* of each panel and the scale bars correspond to 10 μm. Images have been extracted from Supplementary Movie 5 and are shown at 5-min intervals. Note also the mobile, punctate G3BP-GFP-containing structures inside NBs

Depending on their physicochemical properties, proteins partition preferentially either in the cytosolic phase or the NB liquid phase. Therefore, phase separation is an efficient process to enrich the viral factories in factors which are required for viral transcription or replication. FRAP experiments revealed that P, although more concentrated in NBs, is able to shuttle between the cytosol and the NBs. This may help to recruit cellular partners in the viral factories. Indeed, several identified partners of P are recruited inside NBs. This is the case for the focal adhesion kinase FAK[19] and heat shock protein HSP70[18]. All these proteins have been demonstrated to have proviral activities[18, 19] and are also associated with NB-like structures in the minimal system (Supplementary Fig. 8A).

On the other hand, phase separation may also exclude proteins with antiviral properties. Particularly, we show that SGs, which are liquid organelles containing intracellular pattern recognition receptor acting as sensors of RNA virus replication[20, 40, 41], do not fuse with NBs (Fig. 3). This indicates that SGs and NBs are forming non miscible liquid phases. Immiscibility of cellular liquid phases has already been observed and underlies the formation of nucleolar subcompartments[42]. It is worth noting that Rhabdoviruses may have evolved different ways of dealing with SGs: VSV infection also induces the formation of SG-like structures which, however, colocalize with viral replication proteins and RNA[43].

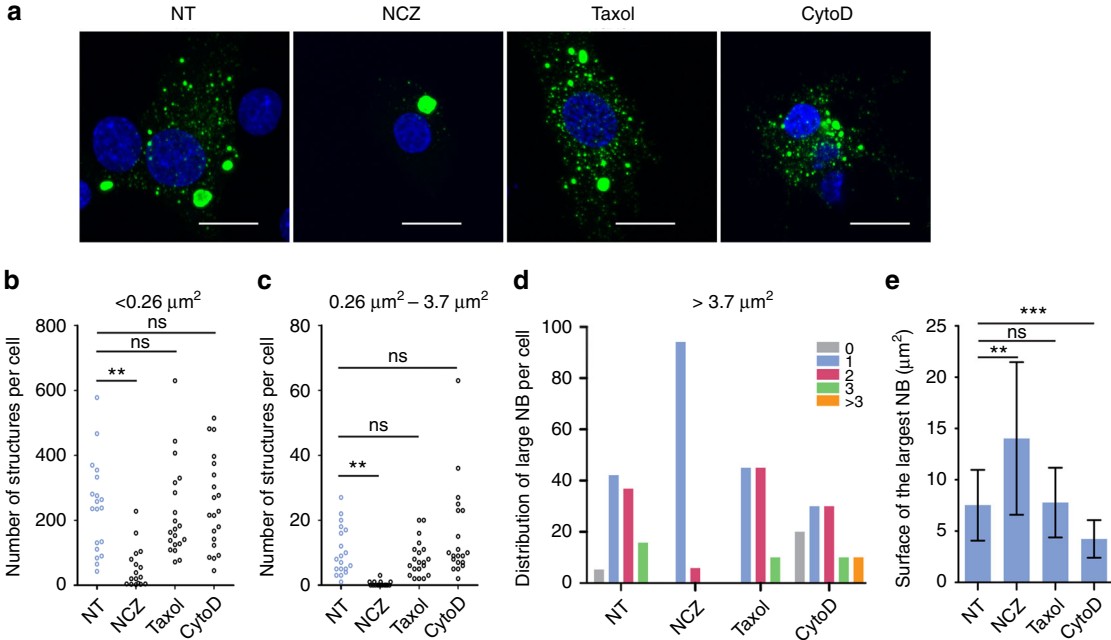

**Fig. 4** Effect of cytoskeleton-disrupting drugs on formation and evolution of NBs. Nocodazole (NCZ, 2 μM), Taxol (1.25 nM) and Cytochalasin D (Cyto D, 2.5 μM) were added 1 h before and kept all along infection. BSR cells were infected with CVS strain at a MOI of 0.5 and fixed at 16 h p.i. NT: non treated cells. **a** Confocal analysis was performed after staining with a mouse monoclonal anti-N antibody followed by incubation with Alexa-488 donkey anti-mouse IgG. DAPI was used to stain the nuclei. Scale bars correspond to 15 μm. **b–d** Quantification of cytoplasmic structures labeled with anti-N antibody in treated and non-treated cells. The number of small dots (surface < 0.26 μm$^2$) **b**, of intermediate inclusions (surface between 0.26 and 3.7 μm$^2$) **c**, and of large inclusions (surface >3.7 μm$^2$) **d** per cell was quantified using the image toolbox of MatLab software as described in the experimental procedures. **p < 0.01, ns: not significant (n = 20, two tailed Mann Whitney U test). **e** Average surface of the largest inclusion in the cell 16 h p.i. in non-treated and treated cells. Areas of inclusions were determined using the image toolbox of MatLab software as described in the experimental procedures. The mean is shown with error bars representing the SD. ns: not significant; **p < 2.10$^{-3}$; ***p < 10$^{-3}$ (n = 20, two tailed Welch's t-test)

In the presence of Nocodazole, which depolymerizes microtubules, the transport of RNPs inside the cytosol is inhibited (Fig. 5b,c). In agreement with this result, live cell imaging revealed a tight association between mobile RNPs and microtubules (Fig. 5d). This explains the previously observed inhibitory effect of Nocodazole on viral production[11]. On the other hand, the ejection of RNPs from NBs is not affected by Nocodazole and, therefore, is not microtubule-dependent. An attractive hypothesis is that a conformational change of RNPs (e.g., their compaction) modifies their physicochemical properties and drastically decreases their solubility in the NB liquid phase, which results in their ejection. Remarkably, no material ejection from NB-like structure is observed in the minimal system. Therefore, the trigger for ejection is only present upon RABV infection.

As depicted in the model presented in Fig. 7, ejected RNPs are transported in the cytosol far away from the NBs. Then, they can either be incorporated into new virions or give rise to new viral factories. This explains the burst of NBs of intermediate size after 12 h of infection. In presence of Nocodazole (Fig. 7), ejected RNPs remain in the vicinity of the NB. They form new viral factories which fuse with the NB. This explains why, in the presence of the drug, a large single NB is generally observed (Fig. 4).

Cytochalasin D, which depolymerizes actin filaments, has little effect on the number of inclusions and free RNPs formed all along the viral cycle. However, our data indicate a decrease of the size of the largest NBs (Fig. 4). This suggests that the actin filament network limits the fragmentation of NBs.

An overview of the literature suggests that the liquid nature of viral factories might be generalized to several other negative RNA viruses (either segmented or not). This is exemplified by FRAP experiments performed on fluorescent Borna Virus phosphoprotein which is also found in spherical inclusions in the nucleus

of infected cells[44]. This is also suggested by data obtained (i) on filoviruses which forms spherical perinuclear inclusions which are viral replication centers[12, 45], (ii) on several paramyxoviruses and pneumoviruses of which N and P proteins form spherical inclusions during the infection[46–48] and (iii) on bunyaviruses of which genomes segments are found 'aggregated' in spherical structures[49]. Therefore, it appears that the formation of such liquid viral factories constitutes a signature of negative RNA viral infection. It is thus highly probable that the cell has developed mechanisms sensing the presence of such structures and that viruses have in turn developed strategies to increase the furtiveness of their factories. Finally, the characterization of the physicochemical nature of those liquid viral factories will pave the way toward innovative antiviral strategies.

## Methods

**Cells**. N2A cells (mouse neuroblastoma, ATCC reference: CCL 131), U373-MG cells (human gliobastoma astrocytoma, ATCC reference: HTB17) were purchased from the ATTC organization (http://www.lgcstandards-atcc.org). BSR cells, a clone from BHK 21 (Baby Hamster Kidney) were obtained from Dr Anne Flamand (from the former Laboratoire de Génétique des Virus, Gif, France). All these cells were grown in Dulbeco's modified eagle medium (DMEM) supplemented with 10% fetal calf serum (FCS).

BSR-T7/5, a clone of BHK 21 constitutively expressing the T7 RNA polymerase[50], were grown in DMEM supplemented with 10% FCS, supplemented with 2% Geneticin (Gibco-Life Technologies).

**Viruses**. The challenge virus standard (CVS, French CVS 11[51]) strain of rabies virus was grown in BSR cells. rCVSN2C-P-mCherry virus encoding a P protein C-terminally fused to the fluorescent protein mCherry has been previously described[20] and was grown in N2A cells.

**Antibodies and drugs**. The rabbit polyclonal anti-P antibody was previously described[11]. Mouse monoclonal anti-N antibody (81C4), was produced in a mouse

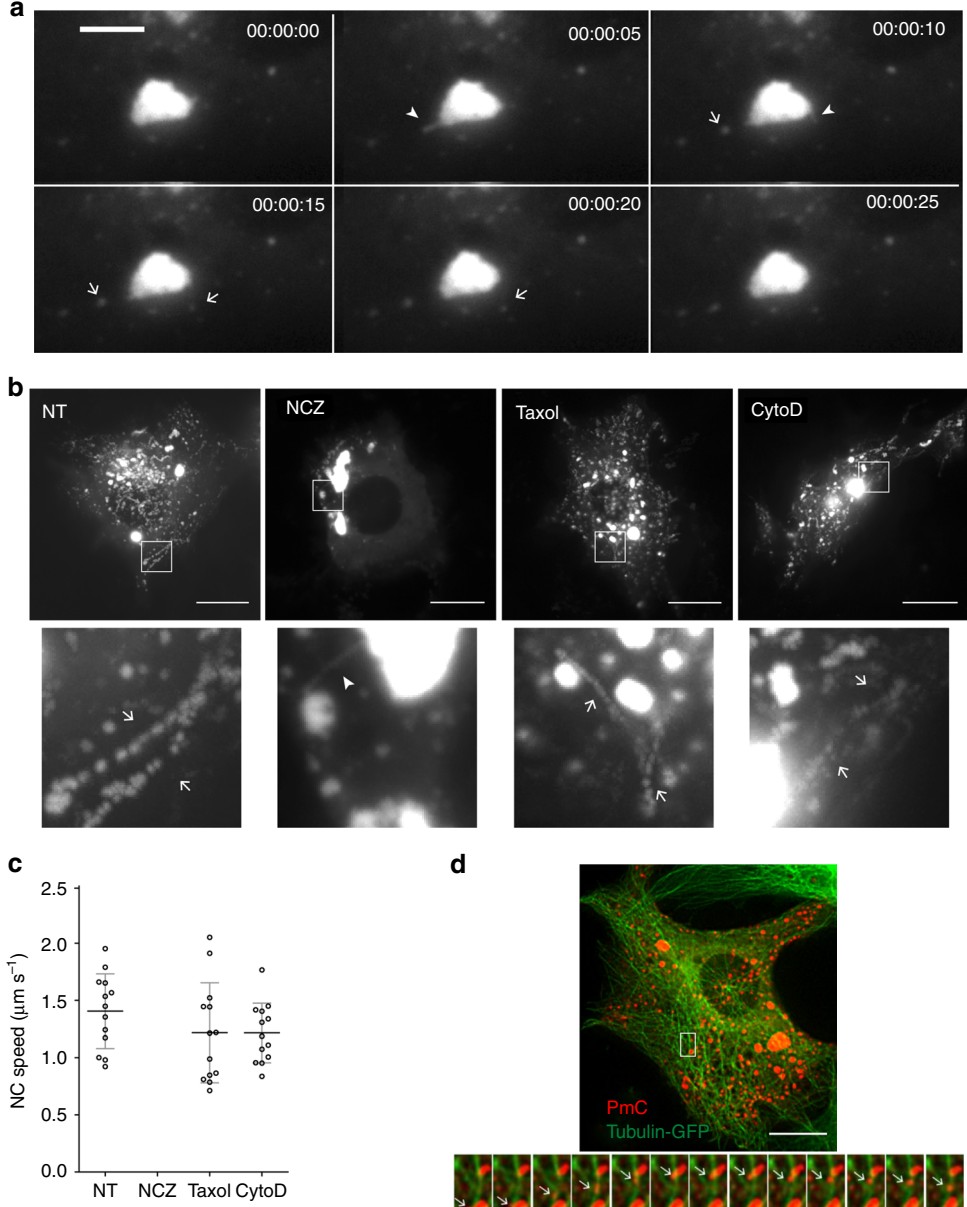

**Fig. 5** RNPs are ejected from NBs and transported along the microtubule network. Live cell imaging was performed on BSR cells infected by rCVSN2C-P-mCherry at a MOI of 0.5 at 16 h p.i. **a** Ejection of RNPs from NBs. The time is indicated in the *upper right corner*. Images have been extracted from Supplementary Movie 6 and are shown at 5-sec intervals. Ejection events are observed on a single image and indicated by arrowheads. The resulting RNPs are indicated by arrows. Scale bar corresponds to 3 μm. **b** Impact of cytoskeleton-disrupting drugs on RNPs transport in the cytosol. Nocodazole (NCZ, 2 μM), Taxol (1.25 nM) and Cytochalasin D (Cyto D, 2.5 μM) were added 1 h before and kept all along infection (NT: non-treated cells). 120 frames (one frame per 1 s, reflecting 120 s) of time-lapse microscopy (such as Supplementary Movie 7) are displayed as maximal intensity projection in order to visualize RNP trajectories which are indicated by arrows in the magnification shown in the lower row. In the NCZ-treated cell (Supplementary Movie 8), an ejection event is indicated by an arrowhead showing that RNP ejection from NBs occurs in absence of an intact microtubule network. Scale bars correspond to 15 μm. **c** Velocity of RNPs in the cytosol in non-treated and treated cells. The speed and the trajectories of RNPs were determined as described in the experimental procedures. Only the RNPs that were tracked on four consecutives images were taken into account. The mean is shown with error bars representing the SD. **d** RNPs are transported along microtubules. BSR cells were co-infected with rCVS N2C-P-mCherry and a modified baculovirus encoding human tubulin-GFP (Cell-light Tubulin-GFP). Images were deconvoluated using the Huygens Imaging software (Supplementary Fig. 5). Scale bar corresponds to 15 μm. Images have been extracted from Supplementary Movie 9 and are shown at 2.5-s intervals (*bottom row*)

immunized with purified nucleocapsids. Both were used at a 1/1000 dilution. Anti-HSP70 MAb (SPA-810) (1/200 dilution) was obtained from Stressgen. Nocodazole (M1404), Taxol (Paclitaxel T1402), Cytochalasin, D (30385), were purchased from Sigma.

**Plasmids**. The plasmid encoding G3BP-EGFP[52] was kindly provided by Dr R Lloyd (Department of Molecular Virology and Microbiology, Baylor College of

Medicine, Houston, TX 77584, USA). The plasmid pFAK-GFP encoding FAK in fusion with GFP has been described previously[19].

pTit-P-mCherry plasmid encoding the P protein of CVS C-terminally fused to the mCherry fluorescent protein has been constructed using Gibson assembly kit (New England Biolabs). The P gene fused to the mCherry has been PCR amplified from the full-length genomic plasmid prN2C-P-mCherry[20].

Plasmids pTit-PΔ, encoding truncated forms of P protein, have been constructed using Gibson assembly kit. PCR products encoding fragments of the

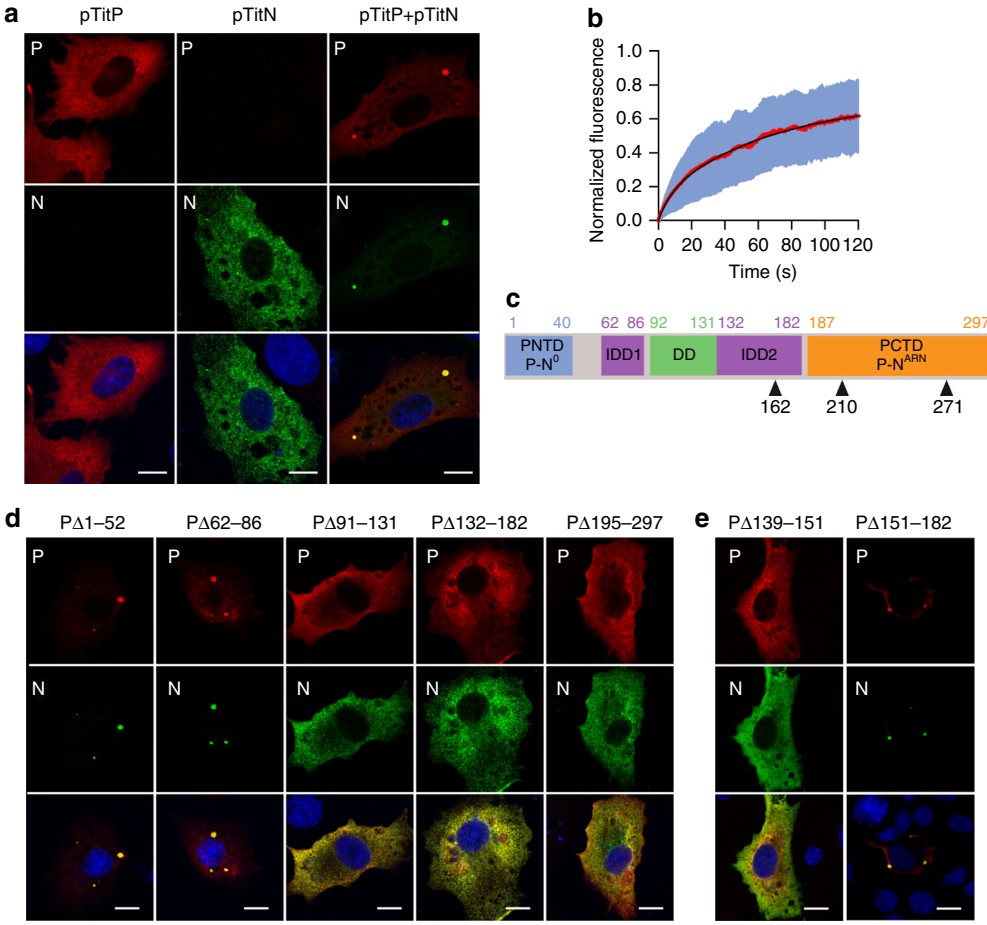

**Fig. 6** Co-expression of N and P leads to the formation of inclusion bodies recapitulating NB properties. **a** BSR-T7/5 cells were co-transfected for 24 h with plasmids pTit-P and pTit-N (in equimolar concentration). N was revealed with a mouse monoclonal anti-N antibody followed by incubation with Alexa-488 donkey anti-mouse IgG and P was revealed with a rabbit polyclonal anti-P antibody followed by incubation with Alexa-568 donkey anti-rabbit IgG. DAPI was used to stain the nuclei. Scale bars correspond to 15 μm. **b** Fluorescence recovery after photobleaching (FRAP) of P-mCherry localized in inclusion bodies in BSR-T7/5 co-expressing P-mCherry and N. The diameter of the photobleached regions was 2.7 μm. FRAP data were corrected for background, normalized to the minimum and maximum intensity, and the mean is shown with error bars representing the SD. Experimental curves were fitted with a two-exponential model (in black). Data were from 21 FRAP events (Supplementary Fig. 6). **c** Domain organization of RABV P polypeptide chain. P contains an N-terminal domain which binds to $N^0$ (PNTD:P-$N^0$), two intrinsically disordered domains (IDD1 and IDD2), a dimerization domain (DD) and a C-terminal domain which binds to RNA-associated N protein (PCTD:P-$N^{ARN}$). Phosphorylation sites in position 162, 210 and 271 are indicated. **d, e** Identification of the P domains involved in inclusion bodies formation. BSR-T7/5 cells were co-transfected with plasmids pTit-N and the indicated construction of pTit-P. N was revealed with a mouse monoclonal anti-N antibody followed by incubation with Alexa-488 donkey anti-mouse IgG and P was revealed with a rabbit polyclonal anti-P antibody followed by incubation with Alexa-568 donkey anti-rabbit IgG. DAPI was used to stain the nuclei. Scale bars correspond to 15 μm

P gene, with overlapping sequences, were assembled using Gibson assembly kit. The plasmids pTit-P S162A, pTit P S162D, pTit S210A, pTit P S210D and pTit P S271A, pTit P S271D were constructed by using QuikChange Site-Directed Mutagenesis Kit (Agilent Technologies).

Plasmids pTit-M, encoding the M protein of CVS and pTit-M-GFP, encoding M fused to the GFP fluorescent protein, have been constructed using Gibson assembly kit. In the latter construct, GFP was inserted between the N-terminal disordered domain and the globular C-terminal part of the M sequence (after residue 26).

In pTit plasmids[53], the gene of interest is under the dependence of the T7 polymerase promoter and the corresponding transcripts contain an internal ribosomal entry site (IRES) located upstream the open reading frame.

**Construction of Recombinant Virus rCVSN2CΔG-P-mCherry.** The full-length recombinant rCVSN2C-P-mCherry infectious clone was described previously[20]. The G coding sequence was removed from the genomic plasmid. The original full-length genomic plasmid was digested with SpeI and MluI restriction enzymes. Two overlapping fragments were amplified by PCR. The first one going from the SpeI site in the P gene to the beginning of the G coding sequence and the second one going from the end of the G coding sequence to the MluI site in the L gene. The PCR products and the digested plasmid were assembled using Gibson

Assembly kit (New England Biolabs) to obtain the resulting plasmid, prN2C-P-mCherry-ΔG.

Recombinant virus (rCVSN2CΔG-P-mCherry) was recovered, as described previously with modifications[19, 54]. Briefly, N2A cells ($10^6$ cells) were transfected using Lipofectamine 2000 (Invitrogen) with 0.85 μg of full-length prN2C-P-mCherry-ΔG, in addition to 0.4 μg pTit-N, 0.2 μg pTit-P and 0.2 μg pTit-L[53], which encode respectively the N, P and L proteins of rabies virus strain SAD-L16. These plasmids were cotransfected with 0.25 μg of a plasmid encoding the T7 RNA polymerase and the plasmid pCAGGS-G[55] encoding the G protein of rabies virus strain Pasteur Virus (PV). Six days post-transfection, the supernatant was passaged on fresh N2A cells transfected with pCAGGs-G, and infectious recombinant viruses were detected 3 days later by the fluorescence of the P-mCherry protein.

**Treatments of cells with drugs.** Cells were kept in Dulbecco modified Eagle medium containing 2 μM Nocodazole, 1.25 nM Taxol, 2.5 μM Cytochalasin D for 1 h before virus inoculation and during infection.

**Immunofluorescence staining and confocal microscopy.** Cells were fixed for 15 min with 4% paraformaldehyde (PFA) and permeabilized for 5 min with 0.1% TX-100 in PBS. Cells were incubated with the indicated primary antibodies for 1 h at RT, washed and incubated for 1 h with Alexa fluor conjugated secondary

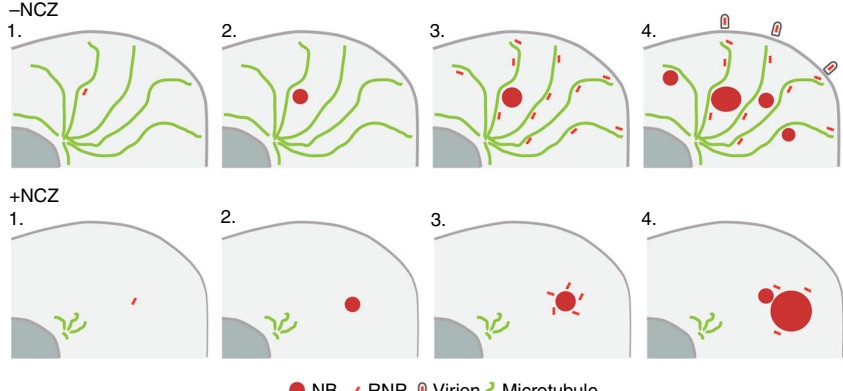

● NB　　✦ RNP　　▯ Virion　　⌇ Microtubule

**Fig. 7** A model for the dynamics of RNPs and NBs in RABV infected cells treated (+NCZ) or not (−NCZ) by Nocodazole. 1. and 2. The initial NB is formed around an incoming RNP. 3. RNPs are ejected from NB by a process which is microtubule independent and are transported away from the initial NB along the microtubule network (−NCZ) or remain in the vicinity of the initial NB when the cells are treated by Nocodazole (+NCZ). 4. In untreated cells (−NCZ), the newly formed RNPs can give rise either to new virions upon budding at the cell membrane or to new viral factories which form NBs of intermediate size. In Nocodazole-treated cells (+NCZ), the newly formed viral factories are located in the vicinity and rapidly fuse with the initial NB which then becomes much larger

antibodies (Thermo Fisher Scientific). Following washing, cells were mounted with Vectashield (Vector labs) containing DAPI. Images were captured using a Leica SP8 confocal microscope (63× oil-immersion objective).

**Quantification of rabies induced structures**. BSR cells were infected with CVS virus at an MOI of 0.5. Cells were fixed at different time post-infection and immuno-stained with a mouse monoclonal anti-N antibody (81C4). For each cell, four planes were acquired by confocal microscopy and a max intensity Z-projection was done using ImageJ. Quantifications were performed using the image toolbox of MatLab software. Briefly, the grayscale image was converted to a binary image. The connected components (CC) of the output black and white image were then identified. For each CC, its area (i.e., the number of pixels) and its eccentricity $e$ (i.e., that of the ellipse that has the same second-moments as the CC) were calculated using MatLab software. This allows the classification of the fluorescent structures formed in the cytoplasm based on their size, and the determination of the area of the largest NB in the cell. This also allows the calculation of the axial ratio $a/b$ (where $a$ and $b$ are the long and short axes of the ellipse) of each inclusion using the formula: $a/b = 1/(1-e^2)^{1/2}$.

**Live cell microscopy**. For live-cell imaging, BSR cells were seeded onto 35-mm μdishes (Ibidi) 24 h before infection. Cells were infected with CVS-N2C-PmCherry rabies virus in DMEM fluorobrite medium (Invitrogen) supplemented with 5% FCS. Live-cell time-lapse experiments were recovered with a Zeiss AxioObserver epifluorescence microscope (63× oil-immersion objective). Cells were maintained at 37 °C and 5% $CO_2$ during imaging.

For live imaging of stress granules, U373-MG cells were transfected using Lipofectamine 2000 (Invitrogen) with a plasmid encoding a G3BP-GFP fusion protein prior to cell infection as previously described[20].

**Quantification of RNPs velocity**. The speed and the trajectories of RNPs were determined manually using the "manual tracking" plugin of the ImageJ software. Only the RNPs that were tracked on four consecutives images were taken into account. The distance between two consecutive positions of a given RNP was measured, allowing the calculation of its instantaneous velocity. Instantaneous velocities along an RNP trajectory were averaged to give the RNP velocity.

**Labeling of tubulin in live cells**. Cell-light Tubulin-GFP (Thermo Fisher Scientific) was used to label tubulin with green fluorescent protein (GFP) in live infected cells. Cell-light Tubulin-GFP is a modified baculovirus that encodes the human tubulin gene fused to the GFP. BSR cells were co-infected with CVS-PmCherry and Cell-light Tubulin-GFP. Live-cell time-lapse experiments were recovered with a Zeiss AxioObserver epifluorescence microscope (63× oil-immersion objective). Images were deconvoluated using the Huygens Imaging software (Scientific Volume Imaging). A blind deconvolution algorithm has been used.

**Electron microscopy**. Infected BSR cells were fixed for 1 h at room temperature in 0.1 M cacodylate buffer (pH 7.2) containing 1% PFA, 2.5% glutaraldehyde and 1% tannic acid. Cells were then post-fixed for 1 h at 4 °C in 0.1 M cacodylate buffer (pH 7.2) containing 1% osmium and 0.8% potassium ferrycianide and stained for 1 h at 4 °C with 2% uranyl acetate in water. Cells were then dehydrated in increasing concentration of acetone and embedded by Epon with 2,4,6-tris

(dimethylaminomethyl) phenol. Polymerization was carried out for 48 h at 60 °C. Ultrathin sections of Epon-embedded material were collected on copper palladium grids (200 mesh). During all the process, cells were kept as a monolayer. The cutting plane is parallel to the cell culture support. The first sections correspond to the basal plane of the cells. These sections were stained with lead citrate and uranyl acetate. Sections were examined with a MET Jeol 1400 electron microscope operated at 80 kV.

**FRAP**. FRAP experiments were performed both on BSR cells infected with recombinant virus rCVSN2CΔG-P-mCherry at 16 h p.i. and on BSR-T7/5 cells co-transfected with pTit-N and pTit-P-mCherry at 24 h post-transfection. Data acquisitions were performed on an inverted Nikon Ti Eclipse E microscope coupled with a Spinning Disk (Yokogawa CSU-X1-A1) and cage incubator to control both temperature (37 °C) and $CO_2$ concentration (5%). After excitation with a 561 nm laser (Cobolt Jive, 150 mW), fluorescence from mCherry was detected with a 100× oil immersion objective (Apochromat NA 1.49), a bandpass filter (607/36 Semrock), and a sCMOS camera (Hamamatsu, Orca-Flash4.0 LT). All the FRAP experiments were performed in identical conditions using iLas FRAP module (Roper Scientific): 5 s prebleach, 40 ms bleach, 120 s postbleach at a frame rate of 1 image every 500 ms. Bleaching was performed in a circular region (diameter 2.71 μm) located at the center of the field of view, with the 516 nm laser at 100%.

For the FRAP data analysis the mean intensity of every bleached region was measured and the recovery signal was normalized to the average of the prebleach signal and also corrected for the bleaching during post bleach imaging[56]. For this double normalization process, the background intensity was estimated by measuring a region outside the cell. To measure the bleaching of the sample during post bleach imaging, the whole cell fluorescence intensity was used for the FRAP experiments of the cytosol while a mean bleaching curve was used for the FRAP experiments of the NBs. To generate this mean bleaching curve, we realized experiments in the same conditions as for the FRAP but without the bleaching event and the mean bleach profile of 18 NBs gave a more robust correction than whole cell measurements due to the very weak numbers of NBs per cell and the strong impact of Z-axis drift of any surrounding NB in these conditions.

After this normalization, individual FRAP curves were averaged to obtain a mean curve. As a single exponential function did not give a satisfactory fit, the mean curve was fitted by a double exponential function:

$$y(t) = y_0 + A_{fast}\left(1 - e^{-k_{fast}t}\right) + A_{slow}\left(1 - e^{-k_{slow}t}\right)$$

The diffusion coefficient $D$ associated with the fast phase was calculated using the following formula[27]:

$$D = \frac{1}{t_{\frac{1}{2}}}\left(\frac{x}{2\text{Erfc}^{-1}(0.5)}\right)^2$$

where Erfc is the complementary error function, $\text{Erfc}^{-1}(0.5) \sim 0,4769$, $x$ is the radius of the beam and $t_{\frac{1}{2}} = \frac{ln2}{k_{fast}}$.

**Statistical analysis**. All numerical data were calculated and plotted with mean ±SD. Results were analyzed by unpaired two tailed Welch's $t$-test or two tailed Mann Whitney U test. The statistical significances of the differences are indicated.

**Data availability**. The authors declare that the data supporting the findings of this study are available within the article and its Supplementary Information files, or are available from the corresponding authors upon request.

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

## Acknowledgements

This work was supported by grants from the Fondation pour la Recherche Médicale (FRM DEQ20120323711) to Y.G. This work has benefited from the core facilities of Imagerie-Gif, supported by "France-BioImaging" (ANR-10-INBS-04-01). We also thank Abbas Abou-Hamdan for careful reading of the manuscript. We thank Sean Whelan for bringing our attention to the sphericity of Negri bodies and Vincent Rincheval for helpful discussion.

## Author contributions

D.B., J.N., R.L.B. and Y.G.: Conceived and designed the experiments, J.N., R.L.B., Z.L., N.S. carried out the experiments; C.L.-G., D.B., J.N., R.L.B. and Y.G. analyzed the data; D.B. and Y.G. supervised the research and wrote the manuscript. All authors contributed to editing the manuscript.

## Additional information

**Competing interests:** The authors declare no competing financial interests.

