## [Peer Review File · Nature Communications]

Reviewers' comments:

Reviewer #1 (Remarks to the Author):

These authors report that the typical inclusion bodies of rabies virus, also known as Negri bodies (NBs), are liquid-like organelles. As such, NBs would share many properties with a variety of cellular proteinaceous membrane-less organelles (PMLOs), just to mention the cytoplasmic germ granules, stress granules, processing bodies (P granules), and the nuclear PML bodies, Cajal bodies, or nucleoli. As many PMLOs are abundant in both RNA and protein, they are also referred to as ribonucleoprotein (RNP) bodies. They represent highly dynamic assemblages, whose components are in direct contact with the cytoplasm or nucleoplasm. Phase transitions in PMLOs are largely supported by intrinsically disordered protein regions (IDRs), though details of the mechanisms behind are still sparse. By controlling a number of RNA processing reactions PMLOs play a central role in cell metabolism. The biophysics of intracellular phase transitions is possibly also of relevance for protein aggregation pathologies. The present manuscript provides several lines of evidence in support of NBs being PMLOs, in particular by FRAP experiments and live microscopy.

The topic of the manuscript is thus of general interest and up-to date, but cellular liquid bodies are continuously being discovered. Why then would the paper merit the prominence of publication in nature communications? As it stands, it probably does not, because large part of the data provided is known from previous publications. However, this reviewer thinks the manuscript can be easily improved and has some potential, in particular with respect to the simplicity of the system, in which two viral proteins are required and sufficient for phase separation and droplet formation. NBs, or similar inclusion bodies of related Rhabdo-, Paramyxo-, and Filoviruses could thus serve as a superb models for studying PMLO phase transitions.

Specific comments:

1. Much of the manuscript describes data on the behavior of NBs which was known before. Dynamics, size, and distribution of NBs (Fig. 1), and the role of actin and microtubules (e.g. Nocodazole-driven fusion (Fig. 4, model in Fig. 7) have been described before (e.g. ref. #11, Lahaye et al., 2009, Albertini et al., Adv Vir Res. 2011, not cited), although the present data are more clear-cut and comprehensive.

In a very recent paper (Nikolic et al. PLoS Path 2016, ref. #18) the authors have already extensively analyzed stress granules (SGs) and Negri bodies (NBs) in rabies-virus-infected cells. There it is stated: "RABV-induced SGs behave as liquid droplets. They are spherical and they grow over time by fusion events upon contact." More specifically, they have already provided experiments on the non-mixing of NBs and SGs, very similar to the ones provided in the present manuscript in Fig. 3. They had noted: "... some G3BP1-containing foci were enclosed inside the NBs, without any connection visible between these structures and the SGs that were surrounding the NBs. Moreover, these G3BP rich structures corresponded to NB areas where the P protein was less present (Fig 6B), excluding co-localization of both proteins. These results showed that G3BP components were present both around the NBs and in specific areas embedded within the NBs. They also indicated

that SGs and viral factories can be very close but still remain distinct structures." The new evidence for NBs representing liquid droplets are essentially provided in Fig. 2 (fusion of NBs, crossing by vesicles or droplet, FRAP) and supplementary figures or videos. The really interesting observations which could make NBs a model for PMLOs are described in Figs 5 and 6, and this part should be elaborated.

2. Fig. 6: Co-expression of RABV N and P proteins indicates that an intrinsically disordered domain (IDD2) and the dimerization domain (DD) of P are important for NB-like structure formation. Though long known in the field, this minimal system is now becoming very interesting with respect to the mechanism of phase transition. One would therefore wish much more experimentation with this minimal system. PMLOs appear to form in a concentration-dependent manner, as expected for liquid-liquid phase separation. Unfortunately, the stoichiometry of transfections (equimolar?) is not stated in the manuscript. I would suggest experiments involving transfection of variable (incl. limiting) ratios of N and P plasmids, to obtain insight into the concentrations and stoichiometry of the driving force (P or N). In addition, the features of IDD2 could be described and analyzed in some more detail. IDRs of PMLOs often have a biased sequence composition, what about the composition of IDD2, and posttranslational modifications, particularly phosphorylation which adds negative charge on S, T, or Y residues of IDRs, and which has been shown to be important for driving phase separation. Is P protein and particularly IDD2 subject to phosphorylation? In addition: can IDD2 be replaced by another IDR, known to be active in other PMLOs?

3. Fig. 5: Live-cell imaging indicates that viral "nucleocapsids" or small NBs are ejected from larger NBs and transported along microtubules to form either new virions or secondary viral factories. Is this observed after N and P expression as well (see Fig. 6). Late NBs seem to be associated with membranes and viral M protein (refs 13, 15). Transfection experiments including M could clarify whether the M protein is involved in pinching off part of the NBs.

4. TLR3 has been described necessary for viral NB formation, and seems to locate in the center of NBs, providing a crystallization point. Is transfection of TLR3 improving NB-like body formation? HSP70 and FAK are associated with viral NBs. Is this composition reflected by NBs generated in the minimal N/P transfection system?

Reviewer #2 (Remarks to the Author):

In this manuscript by Nikolic et.al., the authors report that a certain type of viral factory represents a phase separated liquid microcompartment, which may be a general feature of viral factories. The authors focus on factories made by a rabies virus ("Negri bodies (NBs)"), and show that over the course of 24 hrs of infection the number and size of NBs increases. The structures are highly spherical, fuse with one another, and exhibit rapid fluorescence recovery after photobleaching, consistent with an intracellular liquid phase. The authors go on to show that NBs are apparently immiscible with stress granules, and their size and number is affected in particular by the microtubule cytoskeleton. Expression of NB proteins

alone can result in the formation of liquid like compartments, and data is presented to suggest disordered domains of these proteins play a role, as has been previously reported for numerous other liquid phase compartments.

Overall this is an interesting study and appears to be technically sound. It thus may be appropriate for publication in Nature Communications. However, I think some things would need to be addressed before publication:

1. It would be helpful to include a schematic showing the geometry of assembly and location of various proteins introduced on page 4. There are many proteins: N, L, P, M and G, and then on page 6 NCs are introduced. Where are all of these in the virus, and where are they in the Negri Bodies? For readers outside of the virus field (as would be the case for Nature Communications), such a schematic would be very helpful (perhaps as a large panel in Fig 1).
2. An argument is made that "spherical bubbles of cytoplasm" cross the NBs, and that this supports their liquid nature, which is immiscible with cytoplasm. I found this to be a stretch. Isn't it more likely that what is observed are vesicles being trafficked through the NBs? If it were a "cytoplasmic bubble", how would it be trafficked so directionally through the NB?
3. On page 7, it states that "For safety reason, we used a G gene deleted recombinant RABV...". Why do safety reasons come into play for FRAP. Why is pseudotyping for safety reasons necessary at this point?
4. On page 7, I found the description of the FRAP results confusing. The paragraph beginning with "FRAP experiments were first performed on BSR..." apparently refers to experiments done in the cytoplasm. But I only learned this by inferring from the figure legend, and the subsequent paragraph.

Minor points:

5. page 3 "vimentine" is mis-spelled
6. In figure 2 E-G, the y-axis should have the same scale to facilitate comparison.
7. Beginning of discussion (and I think elsewhere in manuscript): "we provide evidences that...". That word is typically not made plural by adding an s. So I would change to "We provide evidence that..."
8. Page 14, "Furtivity" is not a real word in English, as far as I know.

Reviewer #1 (Remarks to the Author):

Specific comments:

1. Much of the manuscript describes data on the behavior of NBs which was known before. Dynamics, size, and distribution of NBs (Fig. 1), and the role of actin and microtubules (e.g. Nocodazole-driven fusion (Fig. 4, model in Fig. 7) have been described before (e.g. ref. #11, Lahaye et al., 2009, Albertini et al., Adv Vir Res. 2011, not cited), although the present data are more clear-cut and comprehensive.

In a very recent paper (Nikolic et al. PLoS Path 2016, ref. #18) the authors have already extensively analyzed stress granules (SGs) and Negri bodies (NBs) in rabies-virus-infected cells. There it is stated: “RABV-induced SGs behave as liquid droplets. They are spherical and they grow over time by fusion events upon contact.” More specifically, they have already provided experiments on the non-mixing of NBs and SGs, very similar to the ones provided in the present manuscript in Fig. 3. They had noted: “... some G3BP1-containing foci were enclosed inside the NBs, without any connection visible between these structures and the SGs that were surrounding the NBs. Moreover, these G3BP rich structures corresponded to NB areas where the P protein was less present (Fig 6B), excluding co-localization of both proteins. These results showed that G3BP components were present both around the NBs and in specific areas embedded within the NBs. They also indicated that SGs and viral factories can be very close but still remain distinct structures.” The new evidence for NBs representing liquid droplets are essentially provided in Fig. 2 (fusion of NBs, crossing by vesicles or droplet, FRAP) and supplementary figures or videos. The really interesting observations which could make NBs a model for PMLOs are described in Figs 5 and 6, and this part should be elaborated.

As suggested by the referee, we have characterized more thoroughly the minimal system. Those new data are now presented in a new panel in figure 6 (6E) as well as three new supplemental figures (S6, S8 and S9) and three movies (S7, S11 and S12).

2. Fig. 6: Co-expression of RABV N and P proteins indicates that an intrinsically disordered domain (IDD2) and the dimerization domain (DD) of P are important for NB-like structure formation. Though long known in the field, this minimal system is now becoming very interesting with respect to the mechanism of phase transition. One would therefore wish much more experimentation with this minimal system. PMLOs appear to form in a concentration-dependent manner, as expected for liquid-liquid phase separation. Unfortunately, the stoichiometry of transfections (equimolar?) is not stated in the manuscript. I would suggest experiments involving transfection of variable (incl. limiting) ratios of N and P plasmids, to obtain insight into the concentrations and stoichiometry of the driving force (P or N). In addition, the features of IDD2 could be described and analyzed in some more detail.

As suggested by the referee, we have investigated the dependence of phase transition on the stoichiometry of transfection. This is shown in a new supplemental figure (S6). NB-like structures were observed with ratios of N and P plasmids going from 3:1 to 1:3. In more limiting concentrations of one of the plasmids (ratios 9:1 or 1:9), no NB-like structures were

observed even when both proteins were detected in the cell. This is now indicated in the text on p 10.

IDRs of PMLOs often have a biased sequence composition, what about the composition of IDD2, and posttranslational modifications, particularly phosphorylation which adds negative charge on S, T, or Y residues of IDRs, and which has been shown to be important for driving phase separation. Is P protein and particularly IDD2 subject to phosphorylation?

We have now characterized more thoroughly the region of IDD2 important for driving phase separation. We now show that it corresponds to the amino terminal part of IDD2 (residues 132-150). This is now presented in a new panel of figure 6 (6E) and on the top of page 12. This is discussed on the bottom of p13 and we provide alignment of lyssavirus P amino acid sequences (supplemental figure S10) showing that IDD2 and residues 132-150 are poorly conserved compared to other P domains.

Concerning P phosphorylation, domains IDD2 and PCTD are phosphorylated on serine residues in position 162, 210 and 271 (now indicated on figure 6C). These residues were mutated either into an alanine or into an aspartate residue (to mimic the serine in its phosphorylated state). Those mutations were without effect on the ability of P to induce NB-like structure formation. Those data are presented in Figure S9 and in the result section (bottom of p11).

In addition: can IDD2 be replaced by another IDR, known to be active in other PMLOs?

The only other IDR available in the lab was that of VSV P. The replacement of region 132 to 181 of RABV P by that of VSV P did not result in NB-like structures formation. This only proves that the simple presence of an IDR in this place is not sufficient for liquid phase separation. We decided not to include this negative result (which might depend on the imprecise determination of the boundaries of the domains). Clearly, other IDR has to be tried but we consider that such a systematic analysis is beyond the scope of this manuscript.

3. Fig. 5: Live-cell imaging indicates that viral “nucleocapsids” or small NBs are ejected from larger NBs and transported along microtubules to form either new virions or secondary viral factories. Is this observed after N and P expression as well (see Fig. 6). Late NBs seem to be associated with membranes and viral M protein (refs 13, 15). Transfection experiments including M could clarify whether the M protein is involved in pinching off part of the NBs.

Unlike NBs, N-P inclusions do not eject material. In addition to previous movie S6, we have added two other movies showing efficient ejection of RNPs from NBs in infected cells (movie S7) and absence of ejection in cells transfected by N and P (S10). Co-transfection with a plasmid allowing M expression did not affect this result (movie S11) indicating that M is not involved in pinching off part of the NBs. This is not surprising: budding and pinching events mediated by M require the presence of a membrane and hijacking of the ESCRT machinery. As NB and NB-like structures are devoid of membranes, RNPs ejection is due to another mechanism which remains to be identified.

4. TLR3 has been described necessary for viral NB formation, and seems to locate in the center of NBs, providing a crystallization point. Is transfection of TLR3 improving NB-like body formation? HSP70 and FAK are associated with viral NBs. Is this composition reflected by NBs generated in the minimal N/P transfection system?

We now provide evidence that Hsp70 and FAK are associated with NB like structures in the minimal system (Figure S8 and p11).

However, in our hands, TLR3 is not found associated with NBs. Using commercially available polyclonal antibodies (rabbit anti-TLR3 -ab62566- from Abcam), we observed that TLR3 forms dots in the cytosol which do not co-localize with NBs in infected cells (see figure 1 below). As the previous MAb (ref Sc Q18 from Santacruz) which was used in Ménager et al. PLoS Pathog. 5:e1000315 (2009), is no more commercially available, we have no explanation for this discrepancy.

We would like to point out that TLR3 is membrane associated and, so far, we have no evidences for the presence of membranes in NBs. This is therefore consistent with our observation.

Figure 1: Localisation of TLR3 in infected cells and in cells co-transfected with plasmids pTit-P and pTit-N. BSR T7/5 cells were non infected (top row), infected with RABV (CVS) at a MOI of 1 (medium row) or co-transfected with plasmids pTit-P and pTit-N in equimolar concentration (bottom row). Cells were fixed 16h post infection or 24h post-transfection. Confocal analysis was performed after staining with a mouse anti-N antibody followed by incubation with Alexa-568 donkey anti-mouse antibody, and a rabbit-TLR3 antibody (ab62566, Abcam) followed by incubation with Alexa 488 donkey anti-rabbit IgG.

Reviewer #2 (Remarks to the Author):

1. It would be helpful to include a schematic showing the geometry of assembly and location of various proteins introduced on page 4. There are many proteins: N, L, P, M and G, and then on page 6 NCs are introduced. Where are all of these in the virus, and where are they in the Negri Bodies? For readers outside of the virus field (as would be the case for Nature Communications), such a schematic would be very helpful (perhaps as a large panel in Fig 1).

We agree that our previous description of rabies virion was a bit succinct. We have now detailed a little bit more the viral structure. We do not refer anymore to NCs in the text but more accurately to ribonucleoprotein (RNP). The distinction between the nucleocapsid (N + genomic or antigenomic RNA) and RNPs (nucleocapsid plus L and P) is now explicit in the introduction. We also make a reference to a general review on the subject. We have not drawn a schematic panel: such panels are largely accessible on several websites (e.g. <https://www.cdc.gov/rabies/transmission/virus.html>).

2. An argument is made that "spherical bubbles of cytoplasm" cross the NBs, and that this supports their liquid nature, which is immiscible with cytoplasm. I found this to be a stretch. Isn't it more likely that what is observed are vesicles being trafficked through the NBs? If it were a "cytoplasmic bubble", how would it be trafficked so directionally through the NB?

We fully agree with this remark. We have changed the text in P7 accordingly: "Furthermore, we also sometimes observed spherical bubbles crossing the NBs (Figure 2C, movie S3). The directionality of their movement suggests that they are vesicles being trafficked through the NBs. This reinforces the idea that NBs are made of a fluid phase which can reversibly deform when encountering a physical barrier."

3. On page 7, it states that "For safety reason, we used a G gene deleted recombinant RABV...". Why do safety reasons come into play for FRAP. Why is pseudotyping for safety reasons necessary at this point?

RABV is a pathogen which requires L2/L3 safety level (depending on the strain used). The FRAP apparatus is located in a L1 laboratory. We were therefore obliged to use a pseudotyping strategy to comply with the regulation.

4. On page 7, I found the description of the FRAP results confusing. The paragraph beginning with "FRAP experiments were first performed on BSR..." apparently refers to experiments done in the cytoplasm. But I only learned this by inferring from the figure legend, and the subsequent paragraph.

We have now explicitly mentioned in the text that P-mCherry expressed alone is diffuse in the cytosol.

Minor points:

5. page 3 "vimentine" is mis-spelled

This has been corrected.

6. In figure 2 E-G, the y-axis should have the same scale to facilitate comparison.

The y-axis have now the same scale.

7. Beginning of discussion (and I think elsewhere in manuscript): "we provide evidences that...". That word is typically not made plural by adding an s. So I would change to "We provide evidence that..."

This has been corrected.

8. Page 14, "Furtivity" is not a real word in English, as far as I know.

Furtivity has been replaced by furtiveness.

REVIEWERS' COMMENTS:

Reviewer #1 (Remarks to the Author):

The authors have addressed all major issues raised by this reviewer, and provide novel and satisfactory data regarding the stoichiometry of N and P in droplet formation, co-localization of cell factors in minimal N/P droplets, further characterization of P domains required for ND formation. Although (as always) new questions must arise with new data (e.g. on the stoichiometry where N and P can be limiting) I think the ms now merits publication in Nat. Comm.

Congratulations!
Klaus Conzelmann